# Smartphone-Based Human Sitting Behaviors Recognition Using Inertial Sensor

**DOI:** 10.3390/s21196652

**Published:** 2021-10-07

**Authors:** Vikas Kumar Sinha, Kiran Kumar Patro, Paweł Pławiak, Allam Jaya Prakash

**Affiliations:** 1Department of Electronics and Communication Engineering, National Institute of Technology, Rourkela 769008, India; vikas.sinha73@gmail.com (V.K.S.); allamjayaprakash@gmail.com (A.J.P.); 2Department of Electronics and Communication Engineering, Aditya Institute of Technology and Management (A), Tekkali 532201, India; kirankumarpathro446@gmail.com; 3Department of Computer Science, Faculty of Computer Science and Telecommunications, Cracow University of Technology, Warszawska 24, 31-155 Krakow, Poland; 4Institute of Theoretical and Applied Informatics, Polish Academy of Sciences, Bałtycka 5, 44-100 Gliwice, Poland

**Keywords:** accelerometer, classifiers, CFS, an inertial sensor, gyroscope, human sitting behaviors, magnetometer, PSO, smartphone

## Abstract

At present, people spend most of their time in passive rather than active mode. Sitting with computers for a long time may lead to unhealthy conditions like shoulder pain, numbness, headache, etc. To overcome this problem, human posture should be changed for particular intervals of time. This paper deals with using an inertial sensor built in the smartphone and can be used to overcome the unhealthy human sitting behaviors (HSBs) of the office worker. To monitor, six volunteers are considered within the age band of 26 ± 3 years, out of which four were male and two were female. Here, the inertial sensor is attached to the rear upper trunk of the body, and a dataset is generated for five different activities performed by the subjects while sitting in the chair in the office. Correlation-based feature selection (CFS) technique and particle swarm optimization (PSO) methods are jointly used to select feature vectors. The optimized features are fed to machine learning supervised classifiers such as naive Bayes, SVM, and KNN for recognition. Finally, the SVM classifier achieved 99.90% overall accuracy for different human sitting behaviors using an accelerometer, gyroscope, and magnetometer sensors.

## 1. Introduction

People spend their lives in three modes from childhood to old age: active, sedentary, and non-active. In childhood, they spend much time in active mode while playing outdoor games, dancing, running, jumping, etc. In middle age, people spend a lot of time passively, such as sitting on the chair at the office and home on a sofa or bed while watching television for long periods. In old age, people spend maximum time only in non-active mode due to lower physical energy levels. Today, people spend little time in active mode and maximum time in sleeping and sedentary mode. In a sedentary lifestyle, people spend much time sitting and sleeping because of the short life schedule, and people do not care about sitting postures. Sitting is not an issue, but sitting in wrong postures for a long duration in a daily routine may lead to major health issues.

Nowadays, sitting has become a kind of smoke that can lead to many health issues such as back pain [1], sciatica [2], and cervical spondylosis [3]. A traditional technique to analyze patient sitting postures is to let a patient sit on the chair of the hospital in front of the observer such as doctor or therapist, or nurse and complete question answers regarding sitting postures that has been replaced by a smart cushion system [4]. Generally, the diagnostic process takes approximately an hour, which is too short for the complete observation of the patient’s health status. If major issues are found on the spine of the patient, doctors prefer an X-ray to check the curvature of the spine and can detect the spine and the damaged part in the spine of the human body. However, X-ray, computerized tomography (CT) scan, and magnetic resonance imaging cannot be used regularly. 

In [4,5,6,7,8], a smart chair-based approach is applied to improve the sitting postures. The cushion-based smart chair combines pressure sensors and IMU to monitor the sitting behavior at the workplace, in the car, and wheelchair. A generated dataset of sitting behavior has been featured in approximate entropy and standard deviation to monitor and recognize the activities and activity levels [5]. In [6], a smart cushion system was implemented to monitor the sitting activities containing calibrated e-Textile sensors and developed a dynamic time warping-based algorithm to recognize the human sitting behaviors. Sitting posture monitoring systems (SPMSs) are used to help assess the real-time postures of the person and improve the sitting behaviors [7,8]. Roh et al. [6] proposed a system to monitor six sitting behaviors by mounting only four low-cost load cells onto the chair’s seat plate and one on the backrest. A crucial role is played by ergonomic information for the seated person to improve the sitting behaviors and the attitude of the seated person [9,10,11,12].

The basic descriptions of machine learning classifiers used in this work are given as follows: the KNN is the non-parametric algorithm that is most popular for its simplicity and effectiveness [13]. In the KNN classifier, it does not require a learning process [14]. The KNN method classifies a given object based on the closest training object(s) [15]. The KNN classifier has two open issues to be addressed [16]; the first is to measure the similarity between two points, and the other is to choose an ideal value for K [17]. The authors in [18] presented an embedded system designed to perform a KNN classifier to achieve 75% accuracy by applying condensed nearest neighborhood as a prototype selection technique, data balance with Kennard stone, and PCA to reduce the dimension.

The support vector machine (SVM) [19] is a supervised learning approach that analyzes the dataset and recognizes patterns, mainly used for classification and regression analysis [20]. The mathematical expression of the SVM algorithm is covered in [21,22]. The SVM classifier aims to find the optimal separating hyper-plane between two classes for some labeled training datasets [23]. The SVM classifier can also be used for different gesture recognition, for example SVM classifier used in [24] for hand gesture recognition and referred publicly available dataset. It is also one of the supervised classifier techniques, which is popular for its easy implementation and simplicity. In the naive Bayes classifier, the input feature needs to be independent. In contrast, the conditional likelihood function of each sitting behavior can be expressed as the product of probability density functions [25].

To improve analyzing sitting postures, we need to replace the traditional techniques and think of advanced techniques [26]. As we are living in the 21st century and people are using a smartphone for their convenience. First, we need to define different sitting postures as discussed in previous literature, such as forward position, middle position, or backward position, forward sitting posture, reclined sitting posture, slumped sitting posture, laterally tilted left or right sitting posture, arm back leaning, right over left, twist left, poking chin, arm learning, crossed legs right over left or left over right sitting posture and postures in school children [18,27,28,29,30,31,32,33]. The proposed study covered five different sitting postures, which are illustrated in Figure 1. These sitting postures have been considered major sitting activities by employees in the office chairs from previous studies.

The inertial sensors are inbuilt with a smartphone which consists of an accelerometer, gyroscope, and magnetometer. By using the IMU sensor, the movement of the body can be easily tracked. The IMU sensor is attached to the upper rear trunk to track the exact movement of the spine, which will help recognize the postures of the body [28]. Sitting postures have been categorized into two categories: the first is correct posture and the second incorrect posture. This paper considers one correct, straight movement and four incorrect postures: left, right, front, and back movements. 

Some important related papers are discussed based on used sensors, classifiers, and their accuracy in Table 1. Some of the methods listed in Table 1 have limitations like no. of subjects, no. of postures, complexity, convergence time, cost, and less accuracy. This paper implements a smartphone to detect the time spent on correct and incorrect sitting postures during office working hours. The novel contributions of the paper are as follows:

Sitting behaviors dataset is self-generated in the current paper.
The inertial sensor in a smartphone is helpful in monitoring the sitting behaviors of office workers continuously.Determining the time spent on different sitting behaviors, whether correct or incorrect.The user can detect the correct and incorrect sitting behaviors.


The rest of the paper is organized as follows: Section 2 discusses the framework of the current paper, which is illustrated in Figure 1. Data collection, pre-processing, feature extraction, feature selection, and classifiers are briefly discussed. Section 3 represents the results of the experimental setup and discussion of the results. Section 4 concludes the paper and compares the outcomes.

## 2. Framework of Smartphone-Based Sitting Detection

Many different ways have been discussed in previous studies to monitor the multiple sitting postures in the chair. In the proposed approach, the inertial sensor of the smartphone is used as a sitting behaviors detector. The smartphone is one of the best approaches to monitor daily sitting activities at the home or office. In this study, cost, data access, compatibility, unobtrusive use, and system deployment were prioritized. 

Our system can collect a dataset for the five different static movements of the body while sitting in the chair, as illustrated in Figure 1. The smartphone is attached to the rear upper trunk at second thoracic vertebrae T2 to gather the measurable dataset, as illustrated in Figure 2. The anatomical landmark of the sensor on the human body is shown in Figure 2. All the datasets collected from the inertial measurement unit (IMU), inbuilt in the smartphone, are classified after feature extraction and feature subset selection process. The system can identify the postures when the subject moves from the correct sitting posture to incorrect sitting postures. All collected raw datasets cannot directly be fed to the classifiers. To recognize the sitting postures properly, five steps are required to be followed: the first is raw data collection for different sitting behaviors, the second is data pre-processing, the third is feature extraction, the fourth is feature selection and the fifth step is classification, as shown in Figure 1 and are discussed further in the current paper.

### 2.1. Data Collection and Preprocessing

In this work, a new dataset was generated for detecting human sitting behavior while sitting on a chair. Six subjects (four males and two females) participated in generating the dataset by inertial sensors. The age band of 26 ± 3 years was considered for the study. The IMU sensor was attached to the rear upper trunk of the subject. The IMU of the smartphone was used because it is more user friendly. All of the dataset was generated at a 50 Hz sample rate by the sensors data collector one android application. 

A total of five general movements were considered while sitting on the chair in the office as follows:
Left movement;Right movement;Front movement;Back movement;Straight movement.


All different sitting activities were performed by the subjects for different time intervals. A sample sequence of all different activities performed at the chair is as follows: left movement for 712 s, the right movement for 756 s, the front movement for 665 s, the back movement for 590 s, and straight movement for 549 s. Each activity was performed by the subjects in the presence of an instructor to generate the dataset efficiently. The total number of instances for each activity is reported in Table 2 and analyzed with a pie chart in Figure 3.

#### Hardware Platform

The One Plus 6 smartphone was used as a hardware platform for dataset gathering [37]. It consists of the accelerometer, gyroscope, and magnetometer sensors required to recognize physical activities. They run the Oxygen OS 10.3.4 android operating system. The small, low-power Bosch-BMI160 is a low noise 16-bit IMU designed by Bosch used as an accelerometer and gyroscope at 0.002m/s^2^ and 0.030/rad. The AKM-AK09915 magnetometer is used in One Plus 6 smartphones at 0.6µT resolution. The smartphone is 155.70 mm high, 75.40 mm wide, 7.80 mm deep, and weighs 177 g.

Data pre-processing is a crucial step after data collection before feature extraction. First, we removed the present noise of all collected data and then normalized within the range of −1 to +1 to convert them in the same scale, which will be helpful for better feature extraction from the dataset of different sitting behaviors in the office chair.

### 2.2. Feature Extraction

To calculate feature vectors from the collected dataset, the window of $\omega_{t}$ seconds ($N=\left(\omega_{t}\times f_{s}\right)$ Samples) was considered [13]. Here $f_{s}$ is a sampling frequency of 50 Hz of the dataset and *N* samples. The total acceleration of the accelerometer, gyroscope, and magnetometer can be calculated by [14]:
(1)$$A_{T}=\sqrt{\left({\left(A_{x}\right)}^{2}+{\left(A_{y}\right)}^{2}+{\left(A_{z}\right)}^{2}\right)}$$


*A_x_*, *A_y_*, and *A_z_* are the acceleration along the *x*, *y*, and *z* axes of the accelerometer, gyroscope, and magnetometer. For better understanding, features are divided into two different categories, the first is morphological features, and the second is entropy-based features. Where, morphological features include mean of absolute values [33], harmonic mean [33], variance [37,38], standard deviation [38,39], root mean square and simple squared integral [40], and entropy-based features are wavelet entropy and log energy entropy.

#### 2.2.1. Morphological Features

The morphological features include the study of the morphological features such as structure and shape from the dataset of sitting behaviors which are formulated below:
(2)$$Mean\;Absolute\;Value\;(MAV)=\frac{1}{N}\sum_{i=0}^{N}\left|x_{i}\right|$$
(3)$$S\tan dard\;Deviation\;(\sigma )=\sqrt{\frac{1}{N}\sum_{i=0}^{N}{\left(x_{i}-\mu \right)}^{2}}$$
(4)$$Variance\;(VAR)=\frac{1}{N}{\sum_{i=0}^{N}\left(x-\mu \right)}^{2}$$
(5)$$Skewness=\frac{1}{N}\frac{\sum_{i=1}^{N}{\left(x_{i}-\mu \right)}^{3}}{\sigma }$$
(6)$$Root\;Mean\;Square\;(RMS)=\sqrt{\frac{1}{N}\sum_{i=0}^{N}{\left(x_{i}\right)}^{2}}$$
(7)$$Simple\;Squared\;Integral\;(SSI)=\sum_{i=1}^{N}{\left(x_{i}\right)}^{2}$$
where, μ is mean of dataset, σ is the standard deviation and $x_{i}$ is collected samples. A simple squared integral can help to calculate the energy of the signal [33].

#### 2.2.2. Entropy-Based Features

Two kinds of entropy-based features were introduced: wavelet entropy is the measure of relative energies in the different signals and is used to determine the degree of the disorder [1].
(8)$$Wavelet\;Entropy\;(WE)=-\sum_{i=1}^{N}\left(x_{i}\ln(x_{i})\right)$$


Log energy entropy (LEE): For a time series *x*(*n*) of finite length m, LEE is calculated as [11]
(9)$$Log\;Energy\;Entropy=\sum_{i=1}^{N}\left(\log_{2}{\left(x_{i}\right)}^{2}\right)$$


### 2.3. Feature Subset Selection

The dimension of the features is also the dependent parameter of the performance of the classifiers. One of the best ways to reduce the dimension of the extracted features is by applying a feature selection algorithm [40,41]. In this work, a filter method known as correlation-based feature selection (CFS) is applied as a feature selection algorithm [42,43,44]. CFS quickly identifies relevant features and discards redundant, noisy, and irrelevant features based on appropriate correlation measures. This entire approach enhances the performance of classifiers while also reducing their computing time. Table 3 shows the subset of features depending on their contributions. The CFS technique can anticipate each feature as well as feature redundancy at the same time. Hence, a high correlation of the features of the classes is desirable.

Many different search methods are available to select the feature subset in the CFS technique. In [33], a scatter search technique is used and uses a diversification generation technique for a diverse subset. In the current paper, particle swarm optimization (PSO) search technique was applied among other search techniques. PSO technique was proposed by Eberhart and Kennedy in [45]. PSO-based calibration technique was developed to obtain optimized error parameters such as scale factor, bias, and misalignment errors. The all-required mathematical expression and fitness function are explained in [45,46,47]. By combining the CFS technique and PSO search technique, a total of 27 features were selected out of 85 calculated features shown in Table 3.

### 2.4. Sitting Behavior Recognition Techniques

In this paper, three classifier techniques were applied and compared, the most popular among other recognition techniques such as support vector machine (SVM), K-nearest neighbor (KNN), and naive Bayes after feature selection algorithm to recognize the sitting behaviors. The accuracy of all applied recognition techniques estimation is based on the confusion matrix [47]. True and false predicted values help calculate the accuracy of the sitting behaviors recognition techniques, as shown in Equation (10). In the following paragraph, these all-recognition techniques will be discussed briefly
(10)$$Accuracy=\frac{TP+TN}{TP+TN+FP+FN}$$


## 3. Results and Discussion

In this section, the whole experimental setup and results are discussed. Here, MATLAB R2021a was used to perform all calculations and analyses of the dataset of human sitting behaviors of office workers in the office environment. CFS and PSO are jointly utilized for feature selection among various extracted features to provide the highest performance of the applied classifiers. The inertial sensor unit of the smartphone, which includes an accelerometer, gyroscope and magnetometer sensors, was employed to collect the raw dataset defining the five sitting behaviors of the office workers. The recognition results of each sitting behavior were using 10-fold cross-validation for training. Here we split the dataset in the ratio of 3:1 that is 75% of this dataset is the training cross-validation (around 123,545 instances) dataset and 25% of the dataset is the test set (around 39,956 instances). The feature vectors are the representation of the static sitting movements in *x*, *y*, and *z* directions.

### 3.1. Performance Analysis of Classifiers with Feature Selection of Accelerometer, Gyroscope, and Magnetometer

The performance evaluation of the KNN, naive Bayes, and SVM classifiers for feature selection using an accelerometer, gyroscope, and magnetometer is presented in Table 4. The evaluation is based on the sitting activities with 27 feature subsets extracted using the combination of the accelerometer, gyroscope, and magnetometer sensors. All the classifiers were trained properly with similar training conditions. A multiclass SVM (OAA) with a medium Gaussian (RBF) kernel classifies the test data. In the case of K-NN, the hamming distance metric is chosen to find distance along with the nearest neighbor search method. The comparative analysis of the different K values 3, 5, 7, and 11 of KNN are also illustrated in Table 4. The overall accuracy of KNN (K = 3), KNN (K = 5), KNN (K = 7), KNN (K = 11), SVM and naive Bayes are 99.73%, 99.82%, 99.50%, 99.40%, 99.89%, 97.46%, respectively. From the results, it can be inferred that the KNN at the values of K at 3 and 5 and the SVM classifier outperformed among the other classifiers. The performance of each classifier was evaluated for all the five sitting behaviors, and a comparison of experimental results is shown in Table 4.

The confusion matrix of all five sitting behaviors by the SVM classifier is shown in Table 5. The confusion matrix in Table 5 is formed for the instances of four different sitting behaviors confused with each other and straight movement. In the confusion matrix, each row represents instances in true class, and the column represents instances in predicted class.

### 3.2. Performance Analysis of the Classifiers with Feature Selection of Accelerometer and Gyroscope

Table 6 illustrates the results of classifiers for feature selection using accelerometer and gyroscope sensors. The classifier results were obtained for the classifier techniques that are KNN (K = 3, K = 5, K = 7, K = 11), SVM, and naive Bayes. The naive Bayes classifier performs poorly among the other classifiers with 91.60% overall accuracy. The KNN for K = 5 and 11 performed poorly with 98.00% overall accuracy compared to the KNN with K values as 3 and 7. The KNN with K = 3 has better accuracy of 98.60% among all the other K values of KNN. The SVM classifier achieves the overall best performance by the SVM classifier of 98.80%, and the overall comparison of classifier results with all input postures is shown in Figure 4.

From Figure 4, it can be inferred that for the right movement, the classifiers KNN (K = 3, K = 5, and K = 7), SVM, and naive Bayes classifiers have the maximum accuracy with 99.78%, 99.76%, 99.73%, 99.88%, and 99.37%. The KNN with K = 11 performed best for left movement with 96.95% accuracy.

The confusion matrix for all considered sitting behaviors of office workers by using accelerometer and gyroscope with the SVM classifier is shown in Table 7. The confusion matrix in Table 7 is formed for five different sitting behaviors confused with each other.

### 3.3. Performance Analysis of the Classifiers with Feature Selection of Accelerometer

The result of all the applied classifiers only by using the accelerometer sensor is shown in Table 8. From Table 8, it can be illustrated that the naive Bayes classifier performed with 91.90% least overall accuracy compared to the other classifier techniques. The KNN (K = 3), KNN (K = 5) and KNN (K = 7) performed comparably with 99.70% accuracy. The KNN with K = 11 performed as the best classifier with 99.60% accuracy compared to all the other techniques. The overall accuracy of SVM is 99.5%. The right movement feature achieved attention by naive Bayes and all KNN classifiers. Whereas, left movement is most accurately classified by SVM classifier with 98.94% accuracy. The confusion matrix for all the considered sitting behaviors of office workers using an accelerometer with KNN (K = 3) classifier is shown in Table 8. The confusion matrix in Table 9 is formed for five different sitting behaviors confused with each other.

### 3.4. Analysis of Results

In this work, we employed smartphone technology as the sensor for analysis of the sitting behavior of the test subjects. This work made an effort to analyze the static sitting behavior for left movement, right movement, front movement, back movement, and straight movement. It was observed that the SVM classifier outperformed the other classifiers with feature selection from accelerometer, gyroscope, and magnetometer. The performance of each classifier was evaluated for all five sitting behaviors. Even for the feature selection using accelerometer and gyroscope sensors, the SVM classifier achieved the best performance. All the applied classifiers with only the accelerometer, the naive Bayes, and all KNN classifiers achieved the best performance for the right movement. Whereas, the SVM classifier most accurately classified the left movement. After analyzing the overall results, the SVM classifier performed better accuracy with considerable (stable) performance for detecting all three postures from three sensors. Still, while considering other classifiers, the performance is not that much accurate by considering all the postures.

## 4. Conclusions

In this paper, five general sitting behaviors such as left movement, right movement, front movement, back movement, and straight movement of office workers were recognized by using the inertial sensor inbuilt in the smartphone with the help of machine learning classification techniques. An efficient smartphone-based framework was developed with the following stages of pre-processing, feature extraction, feature selection, and machine learning classification. Popular activity recognition techniques like naive Bayes, SVM, and KNN were performed with experimentation to classify the different sitting behaviors. In this paper, the comparative analysis among the above-discussed activity recognition techniques was performed for five sitting behaviors of the office worker. The performance of the KNN classifier at different values of K such as 3, 5, 7, and 11 were also observed by the impact of the sitting behaviors on the office chair. Finally, 99.90% accuracy was achieved in the current work for all sitting behaviors on the office chair by the SVM classifier. From the experimental results, we hope that it is possible to distinguish between the considered five different sitting behaviors only by using a smartphone with the inertial sensor.

## Figures and Tables

**Figure 1 sensors-21-06652-f001:**
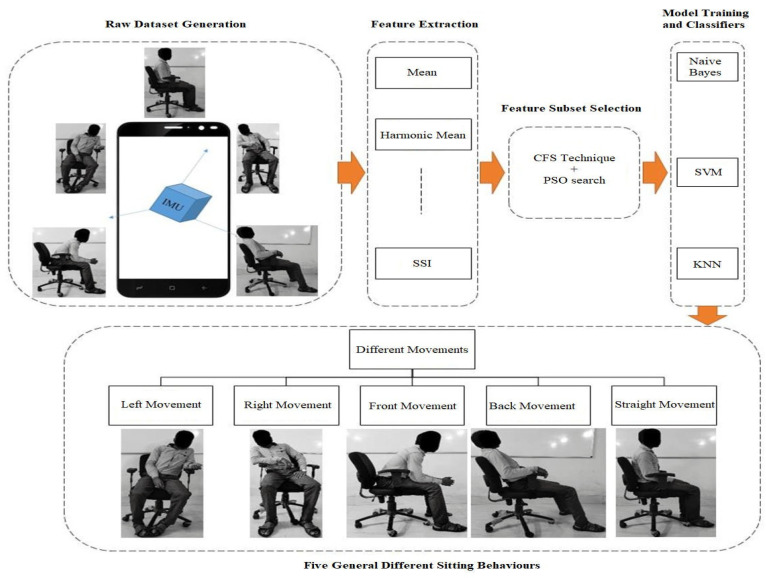
A framework of the proposed system.

**Figure 2 sensors-21-06652-f002:**
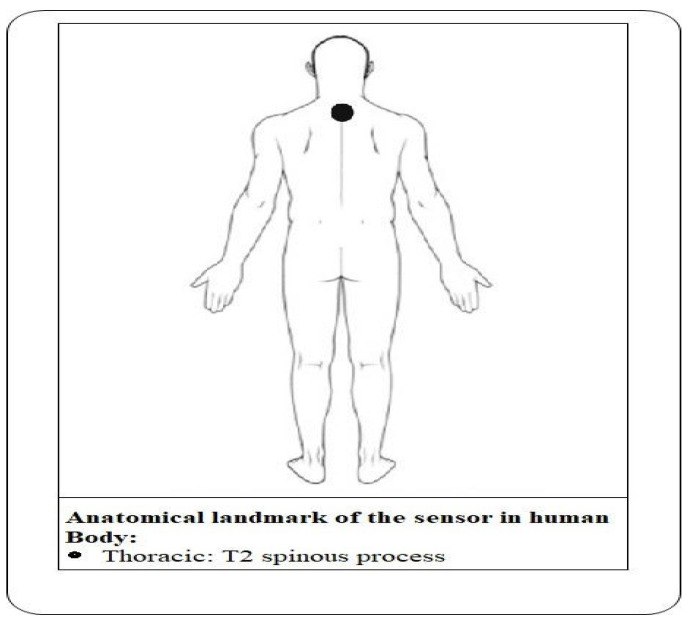
Wearable sensor location in the human body.

**Figure 3 sensors-21-06652-f003:**
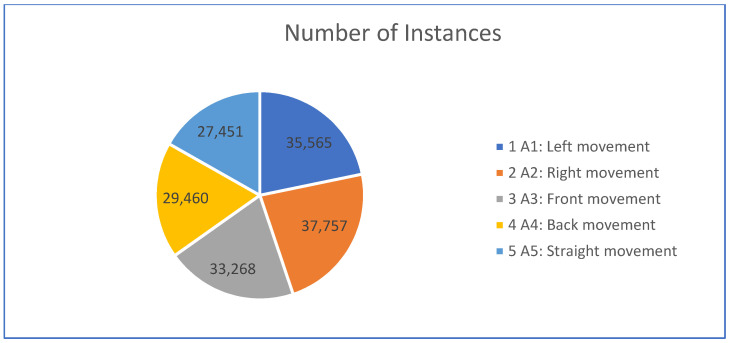
Pie chart representation for a number of instances.

**Figure 4 sensors-21-06652-f004:**
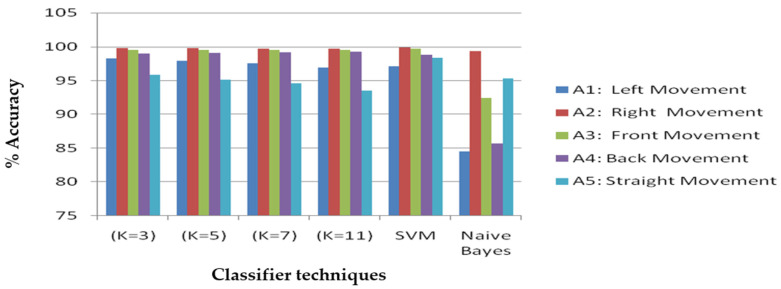
Different classifier results with feature selection of accelerometer and gyroscope.

**Table 1 sensors-21-06652-t001:** Literature review on the basis of previous papers.

S. No	Authors	Type of Sensors	Classifiers	Accuracy (%)	Limitations
1	Xu, Wenyao et al. [4]	Textile sensor array in a smart cushion chair	Naïve Bayes network	85.90	The recognition rate is less.
2	Roh, Jongryun et al. [6]	Low-cost load cells (P0236-I42)	SVM using RBF kernel	97.20	No. of subjects used is less, and power consumption is more.
3	Taieb-Maimon, Meirav et al. [12]	Webcam, Rapid Upper Limb Assessment (RULA) tool.	Sliced inverse regression	86.0	Analyzed only three symptom scales as back symptoms, arm symptoms, and neck pain severity.
4	Arif, Muhammad et al. [33]	Colibri wireless IMU	kNN	97.90	Dataset tested is small, and the optimal set of sensors need to be placed at the appropriate locations on the body.
5	Zdemir et al. [34]	The MTw sensor unit, MTw software development kit	Random forest	90.90	Cost is high, and the convergence time is more.
6	Rosero-Montalvo et al. [18]	Ultrasonic sensor, pressure sensor, Arduinonano, LiPobattery	kNN	75.0	Accuracy reported is much less.
7	Benocci et al. [35]	FSR, digital magnetometer, accelerometer	kNN	92.70	The number of subjects used in the experiment is less.
8	Shumei Zhang et al. [36]	HTC smartphone (HD8282)	kNN	92.70	A posture-aware reminder system can be attached.

**Table 2 sensors-21-06652-t002:** Number of instances per activity.

S. No	Physical Activities	No. of Instances	Time (in Seconds)
1	A1: Left movement	35,565	712
2	A2: Right movement	37,757	756
3	A3: Front movement	33,268	665
4	A4: Back movement	29,460	590
5	A5: Straight movement	27,451	549
	Total	163,501	3272

**Table 3 sensors-21-06652-t003:** Selected features with contribution ratings.

S. No	Selected Features	S. No	Selected Features
1	Total-acceleration	15	Z-magnetometer-SD
2	Total-magnetometer	16	Z-accelerometer-skewness
3	Y-accelerometer-MAV	17	X-gyroscope-skewness
4	X-gyroscope-MAV	18	Y-gyroscope-skewness
5	Y-gyroscope-MAV	19	Z-gyroscope-skewness
6	Y-magnetometer-MAV	20	Y-magnetometer-skewness
7	X-accelerometer-HM	21	Y-accelerometer-LEE
8	X-gyroscope-HM	22	Y-magnetometer-LEE
9	Y-accelerometer-Var	23	X-gyroscope-SSI
10	Z-accelerometer-Var	24	X-accelerometer-WE
11	Z-magnetometer-Var	25	X-gyroscope-WE
12	X-gyroscope-SD	26	X-magnetometer-WE
13	Y-gyroscope-SD	27	Y-magnetometer-WE
14	X-magnetometer-SD		

**Table 4 sensors-21-06652-t004:** Classifier results with feature selection of accelerometer, gyroscope, and magnetometer.

S. No	Activities	KNN (K = 3)	KNN (K = 5)	KNN (K = 7)	KNN (K = 11)	SVM	Naive Bayes
1	A1: Left movement	99.20	99.91	99.91	99.92	99.99	98.51
2	A2: Right movement	99.97	99.97	99.97	99.95	99.98	99.06
3	A3: Front movement	99.96	99.96	99.96	99.94	99.98	99.15
4	A4: Back movement	99.67	99.72	99.70	99.68	99.76	91.89
5	A5: Straight movement	99.89	99.58	98.31	97.68	99.77	98.71

**Table 5 sensors-21-06652-t005:** Confusion matrix of SVM classifier of selected features of accelerometer, gyroscope, and magnetometer.

	S. No	Activity	A1	A2	A3	A4	A5
**True Class**	1	A1	10,669	0	0	0	1
	2	A2	1	11,476	0	0	0
	3	A3	1	0	10,055	1	0
	4	A4	0	0	0	8762	21
	5	A5	0	0	2	16	8044
			**Predicted Class**				

**Table 6 sensors-21-06652-t006:** Classifiers result with feature selection of accelerometer and gyroscope.

S. No	Activities	KNN (K = 3)	KNN (K = 5)	KNN (K = 7)	KNN (K = 11)	SVM	Naive Bayes
1	A1: Left Movement	98.24	97.88	97.56	96.95	97.10	84.55
2	A2: Right Movement	99.78	99.76	99.73	99.69	99.88	99.37
3	A3: Front Movement	99.55	99.53	99.52	99.51	99.66	92.43
4	A4: Back Movement	98.95	99.08	99.17	99.26	98.78	85.66
5	A5: Straight Movement	95.87	95.12	94.57	93.51	98.31	95.31

**Table 7 sensors-21-06652-t007:** Confusion matrix of SVM classifier of selected features of accelerometer and gyroscope.

	S. No	Activity	A1	A2	A3	A4	A5
**True Class**	1	A1	10,341	7	108	8	185
	2	A2	3	11,460	3	2	5
	3	A3	130	2	10,038	1	2
	4	A4	1	1	0	8746	106
	5	A5	69	4	1	59	7868
			**Predicted Class**				

**Table 8 sensors-21-06652-t008:** Classifier results with feature selection of accelerometer.

S. No	Activities	KNN (K = 3)	KNN (K = 5)	KNN (K = 7)	KNN (K = 11)	SVM	Naive Bayes
1	A1: Left movement	99.63	99.57	99.50	99.46	98.94	82.18
2	A2: Right movement	99.95	99.93	99.92	99.91	99.89	99.80
3	A3: Front movement	99.90	99.88	99.87	99.87	99.84	90.62
4	A4: Back movement	99.68	99.72	99.70	99.75	99.60	91.06
5	A5: Straight movement	99.31	99.20	99.15	98.88	99.30	96.28

**Table 9 sensors-21-06652-t009:** Confusion matrix of KNN (K = 3) classifier of selected features of accelerometer.

	S. No	Activity	A1	A2	A3	A4	A5
**True Class**	1	A1	10,634	6	18	1	14
	2	A2	3	11,467	1	0	1
	3	A3	9	0	10,049	0	1
	4	A4	1	0	0	8769	127
	5	A5	15	1	0	39	7994
			**Predicted Class**				

## Data Availability

The data presented in this work are not publicly available due to privacy.

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
