# Peer review of "Smartphone-Based Human Sitting Behaviors Recognition Using Inertial Sensor"

_sensors, 2021, doi:10.3390/s21196652_

Round 1

Reviewer 1 Report

Dear authors, 

I have finished reviewing your article and in my opinion it proposes a useful method to help combat sententarism, which will undoubtedly be a health problem in the short term. In my opinion there are some elements that can be modified so that your work can be used in a better way. 

  1. The parameters with which the classification models were set up should be specified: the metric used in KNN, the kernel chosen for the SVMs and the parameters of the kernel. 
  2. Complement the information in the dataset after the feature selection process, clearly indicating the number of patterns, the classes into which it is divided and the classes per pattern.
  3. In your results you mention the use of 10-Fold Cross-Validation for the training process, please explain if those are the reported results or if the tests were done using a different validation method. 
  4. In the confusion matrix please specify if the rows are the predicted class or the real class, the same with the columns.
  5. Perform a deep review of your paper, there are some typos. 

I hope you find these recommendations useful. 

Author Response

To

Editor-in-Chief

MDPI Sensors

Dear Sir

Herewith I am submitting revised manuscript sensors-1353128, entitled Smart Phone based Human Sitting Behaviors Recognition using Inertial Sensor. We thank the editors, and reviewers for finding the manuscript work ‘interesting contribution in this paper’, ‘The article is at a good level and topic is interesting’. We are also grateful to the reviewers for their careful and thorough evaluation, constructive suggestions and comments. In revising the paper, we have addressed each of the points raised by them and a point to point response is given as follows. We used blue color text in the revised manuscript to highlight the changes made according to the reviewers’ comments.

Thanking you, sir

                                                                                                                                                                                           Yours sincerely

Authors

Enclosures: A copy of comments with Justifications

Comments raised by referee with solved Justifications

(Authors’ Response for the reviewers questions are in bold)

Reviewer-1

Dear authors, 

I have finished reviewing your article and in my opinion it proposes a useful method to help combat sententarism, which will undoubtedly be a health problem in the short term. In my opinion there are some elements that can be modified so that your work can be used in a better way. 

  • The parameters with which the classification models were set up should be specified: the metric used in KNN, the kernel chosen for the SVMs and the parameters of the kernel. 

As per your comment justification is done in the manuscript. In our work, multi class SVM is developed (OAA) with Medium Gaussian (RBF) kernel.

In case of KNN for finding distance hamming distance metric is used with nearest neighbor search (NS) method.

  • Complement the information in the dataset after the feature selection process, clearly indicating the number of patterns, the classes into which it is divided and the classes per pattern.

As per your comment justification is done in the manuscript.

In this work, to classify five different classes of sitting postures (Left Movement, Right Movement, Front Movement, Back Movement, Straight Movement) initially 85 different features are extracted from three sensors. Based on contribution ratings (Correlation measure) only 27 appropriate features are selected for classification.

  • In your results you mention the use of 10-Fold Cross-Validation for the training process, please explain if those are the reported results or if the tests were done using a different validation method. 

As per your comment justification is done.

Yes, in this work for training the data authors are used 10-fold cross validation. Detailed explanation is included in the manuscript. The recognition results of each sitting behavior are using 10-fold cross-validation for training. Here we are splitting the dataset in the ratio of 3:1 that is 75% of this dataset if the training cum cross-validation (Around 1,23,545 instances) dataset and 25% of the dataset is the test set (Around 39,956 instances).

  • In the confusion matrix please specify if the rows are the predicted class or the real class, the same with the columns.

As per your comment justification is done. Generally Confusion Matrix is a table layout that allows visualization of the performance of classification algorithm. Each column of the matrix represents the instances in a predicted class, while each row represents the instances in an actual class.

  • Perform a deep review of your paper, there are some typos. As per your suggestion once again we verified and reviewed the entire manuscript.

Reviewer 2 Report

The authors present an interesting paper on the use of machine learning to analyze sitting behaviors captured via internal sensors within a smart phone.

I did find this work challenging to read, I suggest a thorough scrub of English editing (word choice errors, sentence structure, ect..), and general formatting across the paper.  There are multiple tables of information which could be streamlined with fewer charts while increasing readability. 

Other smaller specific items are noted here:
lines 33-37 - the description here needs reworded, may not be taken without offense.  Additionally, what's the difference between non-active and sedentary? This is not mentioned.  

Table 1 - this should be located prior to using any of the abreviations listed.  This may not be needed is you spell out the abreviation on first use in the paper itself.  That can sometimes be easier on the reader.

lines 72-73 - this sentence needs worked, I'm not sure exactly what is meant to be said here.

Table 2: why do some entries have multiple accuracies listed, it is unclear.  Also, it would help to add another column with a very brief description on the relevant limitation in each method.

Section 2 - why is the T2 vertebrae chosen?  is there relevancy in this or just arbitrary?  Additionally, the time of day, duration, and the population used should be mentioned as part of describing the kind of testing.

line 178 - need to add a reference for the smart phone chosen for readers who are unfamiliar with the specifications.

Section 2.3 - CFS should be described at least briefly since it's at the heart of your method.

Table 4 - the title says selected features with contribution ratings, I'm unclear on what info are the ratings.  Also, this table could save space by using 2-columns.

Section 2.4.1-3 -- these method descriptions should be placed such that the reader reads them prior to mentioning them in your intro.

lines 311-312 -- three of the values you list are equal, but mention that SVM outperforms the rest.  This needs clarification.

Section 3.4 -- looking at the results, you present a suite of classifiers who all do very well except for the one exception.  It would be beneficial for a discussion here on when to chose which classifier for similar situations.  Otherwise, was your intent to show that the multiple KNN perform as well as the SVM?  this is unclear.

My overall recommendation is for the authors to do a thorough scrub of the aforementioned items.  I do feel that this could be a strong paper but it does not present well enough as it is.  

Author Response

To

Editor-in-Chief

MDPI Sensors

Dear Sir

Herewith I am submitting revised manuscript sensors-1353128, entitled Smart Phone based Human Sitting Behaviors Recognition using Inertial Sensor. We thank the editors, and reviewers for finding the manuscript work ‘interesting contribution in this paper’, ‘The article is at a good level and topic is interesting’. We are also grateful to the reviewers for their careful and thorough evaluation, constructive suggestions and comments. In revising the paper, we have addressed each of the points raised by them and a point to point response is given as follows. We used blue color text in the revised manuscript to highlight the changes made according to the reviewers’ comments.

Thanking you, sir

                                                                                                                                  Yours sincerely

Authors

Enclosures: A copy of comments with Justifications

Comments raised by referee with solved Justifications

(Authors’ Response for the reviewers questions are in bold)

Reviewer-2

The authors present an interesting paper on the use of machine learning to analyze sitting behaviors captured via internal sensors within a smart phone.

  • I did find this work challenging to read, I suggest a thorough scrub of English editing (word choice errors, sentence structure, etc..), and general formatting across the paper.  There are multiple tables of information which could be streamlined with fewer charts while increasing readability. 

As per your comment justification is done in the manuscript. All the language related corrections are done and verified with native English speaker.

As per your suggestion required bar and Pie chart (Figure 3 and Figure 5 in the manuscript) are included for better readability.

  • Other smaller specific items are noted here:
    lines 33-37 - the description here needs reworded, may not be taken without offense.  Additionally, what's the difference between non-active and sedentary? This is not mentioned. 

As per your comment justification is done. Modified the description and mentioned few points about non-active and sedentary.

In middle age people spend much time in sedentary mode such as sitting or lying down for long periods. In case of old age due to less energy level they are not doing enough physical activities so that they are in non-active mode.

  • Table 1 - this should be located prior to using any of the abbreviations listed.  This may not be needed is you spell out the abbreviation on first use in the paper itself.  That can sometimes be easier on the reader.

As per your suggestion Table-1 is shifted to prior use of abbreviations listed.

  • Lines 72-73 - this sentence needs worked, I'm not sure exactly what is meant to be said here.

As per your comment justification is done in the manuscript.

  • Table 2: why do some entries have multiple accuracies listed, it is unclear.  Also, it would help to add another column with a very brief description on the relevant limitation in each method.

As per your comment justification is done. Some important related papers have discussed on the basis of used sensors, classifiers and their accuracy in Table 2. Multiple accuracies listed are modified.

As per your suggestion a brief limitations of listed methods are reported in the description and also added a column in Table 2.

  • Section 2 - why is the T2 vertebrae chosen?  is there relevancy in this or just arbitrary?  Additionally, the time of day, duration, and the population used should be mentioned as part of describing the kind of testing.

Generally, T2 vertebrae is the upper middle of the body and also controls the nerves of spine and chest so that it is the better point to choose the collection of data set. Relevant information about data set is already added in Section 2.1.

  • Line 178 - need to add a reference for the smart phone chosen for readers who are unfamiliar with the specifications.

As per your comment justification is done. A suitable reference added for Smart phone One Plus 6 as [48].

  • Section 2.3 - CFS should be described at least briefly since it's at the heart of your method.

As per your comment justification is done. Section 2.3 briefly explained how the features are selected based on contribution rating (Correlation measure) with our own reference [32].

  • Table 4 - the title says selected features with contribution ratings; I'm unclear on what info are the ratings.  Also, this table could save space by using 2-columns.

As per your comment justification is done and Table modified accordingly. Contribution ratings mainly depend on correlation measure and hence high correlated features are considered and discarded redundant, noisy features.

  • Section 2.4.1-3 -- these method descriptions should be placed such that the reader reads them prior to mentioning them in your intro.

As per your suggestion the methods of classification is shifted to section.1.

  • lines 311-312 -- three of the values you list are equal, but mention that SVM outperforms the rest.  This needs clarification.

As per your comment again calculated over all accuracy of all classifiers for all activities from Table 5 and corrected accordingly.

  • Section 3.4 -- looking at the results, you present a suite of classifiers who all do very well except for the one exception.  It would be beneficial for a discussion here on when to chose which classifier for similar situations.  Otherwise, was your intent to show that the multiple KNN perform as well as the SVM?  this is unclear.

In this work, we employed smart phone technology as the sensor for analysis of the static sitting behaviour of the test subjects.  We created own database for five sitting postures such as Left Movement, Right Movement, Front Movement, Back Movement and Straight Movement. Data is collected from smart phone based sensors such as accelerometer, gyroscope and magnetometer. After analyzing the results we conclude that SVM classifier performs better accuracy (Almost 99.90%)with considerable (stable) performance for detecting all postures from 3 sensors but while considering other classifiers the performance is not that much considerable for all the postures (Unstable results for Left movement/Straight movement etc). As per our investigations for some particular postures KNN, Naive Bayes results are inappropriate.

Reviewer 3 Report

The present work deals with the use of inertial sensor which is inbuilt in the smart-phone and can be used to overcome the unhealthy Human Sitting behaviours (HSBs) of the office worker. First of all, I would suggest putting the abbreviations at the end of the paper in a separate section and to avoid using a table. In the introduction, I would introduce the work done in 10.1007/978-3-319-13293-8_6, 10.1109/BHI.2014.6864380 on the promotion of monitoring sitting postures. The methodological approach is quite rigorous, although I would have compared the proposed method with a gold standard approach and discussed it also in the results.

Author Response

To

Editor-in-Chief

MDPI Sensors

Dear Sir

Herewith I am submitting revised manuscript sensors-1353128, entitled Smart Phone based Human Sitting Behaviors Recognition using Inertial Sensor. We thank the editors, and reviewers for finding the manuscript work ‘interesting contribution in this paper’, ‘The article is at a good level and topic is interesting’. We are also grateful to the reviewers for their careful and thorough evaluation, constructive suggestions and comments. In revising the paper, we have addressed each of the points raised by them and a point to point response is given as follows. We used blue color text in the revised manuscript to highlight the changes made according to the reviewers’ comments.

Thanking you, sir

                                                                                                                                                                                           Yours sincerely

Authors

Enclosures: A copy of comments with Justifications

Comments raised by referee with solved Justifications

(Authors’ Response for the reviewers questions are in bold)

Reviewer-3

The present work deals with the use of inertial sensor which is inbuilt in the smart-phone and can be used to overcome the unhealthy Human Sitting behaviour (HSBs) of the office worker. First of all, I would suggest putting the abbreviations at the end of the paper in a separate section and to avoid using a table. In the introduction, I would introduce the work done in 10.1007/978-3-319-13293-8_6, 10.1109/BHI.2014.6864380 on the promotion of monitoring sitting postures. The methodological approach is quite rigorous, although I would have compared the proposed method with a gold standard approach and discussed it also in the results.

As per your comment justification is done in the manuscript.

For better readability the Table-1 is shifted to prior use of abbreviations listed.

As per your suggestion given references are included as [46] and [47]. These references are very much suitable to the work.

The similar methods have discussed on the basis of used sensors, classifiers and their accuracy in Table 2. Some of the techniques listed in the Table-2 are having limitation like complexity, Convergence time, cost and less accuracy. After analyzing the results we conclude that SVM classifier performs better accuracy (Almost 99.90%)with considerable (stable) performance for detecting all postures from 3 sensors but while considering other classifiers the performance is not that much considerable for all the postures (Unstable results for Left movement/Straight movement etc).

Round 2

Reviewer 1 Report

Dear authors,

In my opinion you have addressed al my initial concerns about your work. I have no more concerns or recommendations about your work. 

Reviewer 2 Report

Author's did well incorporating recommended changes, and improved readability significantly.  Suggest Editor to accept in present form.